# Cobalt Minimisation in Violet Co_3_P_2_O_8_ Pigment

**DOI:** 10.3390/ma15031111

**Published:** 2022-01-31

**Authors:** Mª Ángeles Tena, Rafael Mendoza, Camino Trobajo, Santiago García-Granda

**Affiliations:** 1Inorganic Chemistry Area, Inorganic and Organic Chemistry Department, Jaume I University, P.O. Box 224, 12006 Castellón, Spain; 2Physical Chemistry Area, Scientific and Technical Services, Oviedo University-CINN, 33006 Oviedo, Spain; mendozarafael@uniovi.es; 3Inorganic Chemistry Area, Organic and Inorganic Chemistry Department, Oviedo University-CINN, 33006 Oviedo, Spain; ctf@uniovi.es; 4Physical Chemistry Area, Physical and Analytical Chemistry Department, Oviedo University-CINN, 33006 Oviedo, Spain

**Keywords:** minimisation of toxicity, Co_3_P_2_O_8_, α-Mg_3_P_2_O_8_, solid solutions, pigments

## Abstract

This study considers the limitations of cobalt violet orthophosphate, Co_3_P_2_O_8_, in the ceramic industry due to its large amount of cobalt. Mg_x_Co_3−x_P_2_O_8_ (0 ≤ x ≤ 3) solid solutions with the stable Co_3_P_2_O_8_ structure were synthesised via the chemical coprecipitation method. The formation of solid solutions between the isostructural Co_3_P_2_O_8_ and Mg_3_P_2_O_8_ compounds decreased the toxically large amount of cobalt in this inorganic pigment and increased the melting point to a temperature higher than 1200 °C when x ≥ 1.5. Co_3_P_2_O_8_ melted at 1160 °C, and compositions with x ≥ 1.5 were stable between 800 and 1200 °C. The substitution of Co(II) with Mg(II) decreased the toxicity of these materials and decreased their price; hence, the interest of these materials for the ceramic industry is greater. An interesting purple colour with a* = 31.6 and b* = −24.2 was obtained from a powdered Mg_2.5_Co_0.5_P_2_O_8_ composition fired at 1200 °C. It considerably reduced the amount of cobalt, thus improving the colour of the Co_3_P_2_O_8_ pigment (a* = 16.2 and b* = −20.1 at 1000 °C). Co_3_P_2_O_8_ is classified as an inorganic pigment (DCMA-8-11-1), and the solid solutions prepared were also inorganic pigments when unglazed. When introducing 3% of the sample (pigment) together with enamel, spreading the mixture on a ceramic support and calcining the whole in an electric oven, a colour change from violet to blue was observed due to the change in the local environment of Co(II), which could be seen in the UVV spectra of the glazed samples with the displacement of the bands towards higher wavelengths and with the appearance of a new band assigned to tetrahedral Co(II). This blue colour was also obtained with Co_2_SiO_4_, MgCoSiO_4_ or Co_3_P_2_O_8_ pigments containing a greater amount of cobalt.

## 1. Introduction

In the ceramic industry, dark blue is obtained by using compounds or solid solutions containing cobalt. The coordination number of the Co(II) cation is different among the crystalline structures used, and the colouration of powdered compositions changes between blue (CoAl_2_O_4_ with a spinel structure and 88% tetrahedral Co(II), ICSD-260589; Co_x_Zn_2−x_SiO_4_ (0.005 ≤ x ≤ 1) with a willemite structure, and 100% tetrahedral Co(II) ICSD-186367) and violet, blue, or purple (Co_2_SiO_4_ with olivine structure and 100% octahedral Co(II) ICSD-260092; Co_3_P_2_O_8_ with a related olivine structure and 100% octahedral Co(II) ICSD-9850; stable Co_3_P_2_O_8_ with 1/3 octahedral Co(II) and 2/3 Co(II) in C.N. = 5 ICSD-38259) [1].

Cobalt violet orthophosphate, Co_3_P_2_O_8_, is a pigment included in the DCMA Classification of Mixed Metal Oxide Inorganic Colour Pigments (DCMA-8-11-1) [2]. Its use in the ceramic industry is limited because of the large amount of cobalt in this compound. The formation of solid solutions between the isostructural Co_3_P_2_O_8_ and Mg_3_P_2_O_8_ compounds could be used to avoid the toxically large amount of cobalt in this pigment. In compositions that are rich in magnesium, these solid solutions could decrease the amount of cobalt, thus increasing the interest of these materials for the ceramic industry. The substitution of Co(II) with Mg(II) decreases the toxicity of these materials and decreases their price. The Co_3_P_2_O_8_ compound melts at 1160 °C [3,4]. Magnesium orthophosphate melts at 1357 °C [5]. The melting point of some compositions of the solid solutions could be higher than 1200 °C.

The formation of solid solutions through the substitution of ions in a crystalline structure changes the bond strength and modifies the colour of the materials. The colour blue is usually obtained from a tetrahedral CoO_4_ geometry. The colour purple is obtained from LiZn_1−x_Co_x_PO_4_ (0 ≤ x ≤ 0.4) compositions with a LiZnPO_4_ structure, which can be explained by the highly distorted geometry in the CoO_4_ tetrahedra. The shorter Co-O bonds increase the ligand field strength and lead to a blue-shifted absorption, thus developing an excellent purple pigment [6]. The increase in the amount of Co in these compositions decreases the negative b* value of the CIE L* a* b* parameters and the colour change to a violet hue due to the presence of the LiCoPO_4_ phase with Co(II) ions in octahedral coordination together with the LiZnPO_4_ phase with Co(II) ions in tetrahedral coordination [6].

The stable polymorph of the Co_3_P_2_O_8_ compound with the monoclinic Mg_3_P_2_O_8_ structure (α phase) contains Co(II) ions in both a square planar pyramid and an octahedral coordination, Co1 in the 4e site and Co2 in the 2a site (ICSD-38259) [7]. A polymorph of the Mg_3_P_2_O_8_ compound with the Ni_3_P_2_O_8_ structure (β phase) has also been reported (ICSD-9849) [1,8]. The transition temperature from β-Mg_3_P_2_O_8_ to α-Mg_3_P_2_O_8_ (ICSD-261231) is 1055 °C [5]. In the stable Co_3_P_2_O_8_ structure, all of the Co(II) ions are distributed in layers (bc planes), and these layers are joined by PO_4_ tetrahedra. Figure 1 shows two unit cells of the stable Co_3_P_2_O_8_ structure, the projection of nine unit cells in the (001) plane and the details of the oxygens around Co1 (CN = 5) and Co2 (CN = 6), with an edge shared by both polyhedra (two oxygens, O1 and O2, shared between Co1 and Co2). Two Co1-to-Co2 distances—2.896 and 3.165 Å—are shorter than the other Co-to-Co distances (ICSD-38259) [7]. The structure was drawn with the FPStudio program [9,10,11].

Cobalt orthophosphate is soluble in Mg_3_(PO_4_)_2_ at about 1100 K (827 °C) over the whole range of compositions [12]. Information about (Co, Mg)_3_(PO_4_)_2_ solid solutions with the α-Mg_3_P_2_O_8_ structure prepared from mixtures of Co_3_P_2_O_8_ and Mg_3_P_2_O_8_ orthophosphates and fired at 800 °C can be found in the bibliography [13]. The catalytic behaviour of Mg_3−x_Co_x_(PO_4_)_2_ solid solutions (compositions fired at 500–700 °C) shows that the substitution of magnesium with cobalt in the Mg_3_(PO_4_)_2_ structure leads to active and selective phases in the oxidative dehydrogenation of ethane and propane [14]. It seems possible to increase the thermal stability of Co_3_P_2_O_8_ through the formation of these solid solutions with the substitution of Co(II) ions by Mg(II) ions. As far as we know, no information about Mg_x_Co_3−x_P_2_O_8_ solid solutions at T ≥ 1000 °C has been reported.

In Co_2−x_Zn_x_SiO_4_ solid solutions comprising Co_2_SiO_4_ and Zn_2_SiO_4_ compounds with olivine and willemite structures, the intense blue colour is kept with a considerably lower amount of cobalt [15]. In the same way, the colour of some compositions of solid solutions with the structure of cobalt orthophosphate with a small amount of cobalt could also be similar to the colour of the Co_3_P_2_O_8_ compound. It was expected that the change in the coordination number of the Co(II) ion would modify the colouration of the material with respect to that obtained with the olivine and willemite structures.

The aim of this study was the formation of solid solutions from compositions comprising Co_3_P_2_O_8_ and Mg_3_P_2_O_8_ in order to obtain information about the composition and temperature with which the desired colour is developed and to minimise the toxic and expensive amounts of cobalt.

## 2. Experimental Methods

Mg_x_Co_3−x_P_2_O_8_ (0 ≤ x ≤ 3) compositions were synthesised from MgCl_2_·6H_2_O (Scharlau, extra pure), Co(NO_3_)_2_·6H_2_O (Acros Organic, 99%) and H_3_PO_4_ (Merck, 99%) via the chemical coprecipitation method.

Stoichiometric amounts of MgCl_2_·6H_2_O, Co(NO_3_)_2_·6H_2_O and a 0.5 M solution of H_3_PO_4_ in water were added to 100 mL of water. Samples were vigorously stirred for 20 h at room temperature. Then, an aqueous ammonia solution (Panreac, 25%) was added under continuous stirring until reaching pH = 10. The experimental parameters were chosen in order to obtain precipitates of the cations before drying the material. pH = 10 was chosen because, although Co(OH)_2_ precipitates at pH > 7, Mg(OH)_2_ precipitates at pH > 9.5. Under these conditions, the materials were coprecipitated and dried in a stove at 65 °C to evacuate only the water. The Mg:Co:P molar ratio of the starting materials was preserved in this process. The dry samples were fired at 300, 600, 800, 1000 and 1200 °C for 6 h at each temperature.

The development of the crystalline phases at different temperatures was studied by using XRD. The resulting materials were examined using a Panalytical X-ray diffractometer (Malvern Panalytical, Almelo, The Netherlands) with CuK_α_ radiation. A structure profile refinement was carried out using the Rietveld method (Fullprof.2k computer program) [9,10,11]. Diffraction patterns ranging between 6 and 110 (2θ) were collected by employing monochromatic CuK_α_ radiation, a step size of 0.02 (2θ) and a sampling time of 10 s. The unit cell parameters, interatomic distances and Co(II) ion occupation in the two M1 and M2 sites in the stable Co_3_P_2_O_8_ structure were determined in order to investigate the possible formation of solid solutions under these synthesis conditions. The initial structural information was taken from the Inorganic Crystal Structure Database [1].

The Co(II) ion sites and the transfer charge bands in the samples were studied by using UV–vis–NIR spectroscopy (diffuse reflectance). The ultraviolet–visible–near-infrared (UV–vis–NIR) spectra in the range of 200 to 2500 nm were obtained using a Jasco V-670 spectrophotometer and BaSO_4_ as reference substance.

The CIEL*a*b* colour parameters for the fired samples—L* is the lightness axis (black (0) → white (100)), a* is the green (−) → red (+) axis and b* is the blue (−) → yellow (+) axis [16]—were obtained with an X-Rite spectrophotometer (SP60, standard illuminant: D65, an observer of 10°, and a reference sample of MgO).

To test their possible utility in the ceramic industry, the compositions fired at 1200 ºC were enamelled at 3% weight with a commercial glaze (SiO_2_–Al_2_O_3_–PbO–Na_2_O–CaO glaze) onto commercial ceramic biscuits. Many pigments were dissolved in this glaze. The colour of the material was lost when this occurred. Glazed tiles were fired for 15 min at 1065 °C, and subsequently, their UV–vis–NIR spectra and their CIEL*a*b* colour parameters were obtained.

## 3. Results and Discussion

Table 1 shows the evolution of the crystalline phases in the Mg_x_Co_3−x_P_2_O_8_ (0.0 ≤ x ≤ 3.0) compositions according to composition and temperature. A stable Co_3_P_2_O_8_ structure was developed at 800 °C, although small amounts of Mg_2_P_2_O_7_ were also detected when x ≥ 2.0 at this temperature. This crystalline phase was the only crystalline phase detected in all of the compositions at 1000 °C and when x ≥ 1.5 at 1200 °C. Compositions with x < 1.5 melted at 1200 °C, and they could not be removed from the crucible.

The unit cell parameters, volumes and interatomic distances in the stable Co_3_P_2_O_8_ structure (isostructural with the α-Mg_3_P_2_O_8_ structure) were obtained from the Mg_x_Co_3−x_P_2_O_8_ (0 ≤ x ≤ 3) compositions by using Rietveld’s method. Three examples of graphical results are shown in Figure 2. The changes in intensities due to compositional differences can be observed in this figure. Table 2 includes the unit cell parameters and volume, Table 3 includes the M-O (M = Mg, Co) distances, and Table 4 includes the P-O distances. The variations in the unit cell parameters in this structure at temperatures between 800 and 1200 °C are shown in Figure 3 and Figure 4 shows the variations in the M-O and P-O distances at 1000 °C.

For the variations in the unit cell parameters with temperature, only small differences were detected when x > 1.5 (Figure 3). The decrease in the b unit cell parameter with x was consistent with the replacement of the Co(II) ion by the smaller Mg(II) ion (Table 2 and Figure 3). A slight negative departure of Vegard’s law in the b parameter associated with short-range cation ordering [17] can be observed in Figure 3. The small increases obtained in the a and c unit cell parameters indicate a structural distortion produced when Mg_x_Co_3−x_P_2_O_8_ solid solutions were formed with the stable Co_3_P_2_O_8_ structure. This structural distortion is in accordance with the opposite variation in the M1-O distances with the variation in the composition (x) that can be observed in Figure 4. The most remarkable variations were the decrease in M1-O1 (the longest distance) and the increase in M1-O3 with x when x > 1.0. At 1000 °C, the M-O distances were close to the Co-O distances in the Co_3_P_2_O_8_ compound when x ≤ 1.5, and the M-O distances were close to the Mg-O distances in the Mg_3_P_2_O_8_ compound when x > 1.5. When the amount of Mg(II) ions increased, the difference between the shortest and longest distances was smaller than in compositions in which the amount of Co(II) ions was greater. A gradual variation in the M-O distances with x was obtained at 800 and 1000 °C, as well as in the compositions that did not melt at 1200 °C. The great distortion in the position with C.N. = 5 in Co_3_P_2_O_8_ (with Co1-O distances between 1.940 and 2.286 Å according to the ICSD-38259 data) was kept in the compositions in which 0.0 ≤ x ≤ 1.5 at 800 and at 1000 °C. This distortion in the M1 site decreased in accordance with the values of the Mg1-O distances in Mg_3_P_2_O_8_ (with Mg1-O distances between 1.969 and 2.150 Å according to the ICSD-261231 data). The differences in the electron configuration with the seven electrons in the 3d orbitals in the Co(II) ion and with a small orbital penetration effect could explain the fact that the Co-O distance (1.940 Å) was shorter than the Mg-O distance (1.969 Å), although the ionic radius was greater in the Co(II) ion than in the Mg(II) ion. The greater covalence in the Co-O bond than in the Mg-O bond could explain the shorter Co-O distances.

Figure 5 shows the variations in the occupation of Co(II) ions in the 2a (M1) and 4e (M2) sites for the results of the DRX profile refined with the Rietveld method. The initial occupation of each position was calculated by dividing the multiplicity of this position by the multiplicity of the general position in the space group and multiplying that quotient by the value of (3 − x)/3. In accordance with the multiplicity of the sites, a random distribution of Co(II) ions between the M1 and M2 sites corresponded to a ratio of 4/4 = 1 (2/3 of the total Co(II) ions in the M1 site; general 4e position with C.N. = 5) and 2/4 = 0.5 (1/3 of the total Co(II) ions in the M2 site; special 2a position with C.N. = 6). The calculated values were refined, and the results appear in Figure 5 as E1 (experimental occupation in the M1 position) and E2 (experimental occupation in the M2 position). The experimental distribution of Co(II) between the two positions indicates that the Co(II) ion presents a higher preference for the site with C.N. = 5 (4e) than the Mg(II) ion at 800, 1000 and 1200 °C. The octahedral positions (2a) were mostly occupied by Mg(II) ions. This result is in agreement with the literature on compositions at 800 °C [12,13].

Figure 6 shows the UV–vis–NIR spectra of Mg_x_Co_3−x_P_2_O_8_ fired at 800, 1000 and 1200 °C. The three bands at 1100, 580 and 500 nm from the fired Mg_x_Co_3−x_P_2_O_8_ composition were assigned to Co(II) in the octahedral site with Δ/B < 13. These bands could be assigned to the first ^4^T_1_ → ^4^T_2_(F) transition, to the second ^4^T_1_ → ^4^A_2_(F) transition and to the third ^4^T_1_ → ^4^T_1_(P) transition [18]. The bands at 1700–1717, 890 and 480 nm were assigned to Co(II) in a square-planar pyramid coordination (^4^A_2_ → ^4^A_1_(F), ^4^A_2_ → ^4^E(F) and ^4^A_2_ → ^4^E(P) transitions) according to the stable Co_3_P_2_O_8_ solid solutions detected by XRD.

In the spectra at 1000 °C, the absorbance at 1100 nm (the ^4^T_1_ → ^4^T_2_(F) transition was assigned to Co(II) in the octahedral site) was slightly smaller than at 890 nm (the ^4^A_2_ → ^4^E(F) transition was assigned to Co(II) in a square-planar pyramid coordination) when x < 1.0 (small amount of Mg(II) in the compositions), although the most noticeable change was the decrease in absorbance with the composition due to the decrease in the total Co(II) in samples.

The slight changes in the Co-O distances with composition (Figure 4) changed the ligand field strength, and a gradation of purple to violet colour was obtained (Table 5). Purple and violet colours were obtained when the stable Co_3_P_2_O_8_ structure was developed from these compositions. These colourations were kept at 1000 and 1200 °C when x ≥ 1.5, so pigments with a smaller amount of Co(II) were obtained.

According to the CIE L* a* b* parameters (Table 5), an increase in the amount of red colour could be detected starting at 600 °C with a* > 20 when 1.0 ≤ x ≤ 2.5 at 1000 °C. Figure 7 shows the variations in a* and b* with composition (x) at T ≥ 800 °C. All of the compositions with x ≠ 3.0 were violet or purple with a positive a* (red amount) and negative b* (blue amount). The composition with x = 1.5 (Mg_1.5_Co_1.5_P_2_O_8_) showed the greatest amounts of red and blue at 1000 °C, and the composition with x = 2.5 (Mg_2.5_Co_0.5_P_2_O_8_) did so at 1200 °C. The positions of the third transition band of the octahedral Co(II) ion, ^4^T_1_ → ^4^T_1_(P), and the ^4^A_2_ → ^4^E(P) band of the Co(II) ion in a square-planar pyramid coordination could be related to the variation of the amount of red colour (positive a*). The amount of blue colour (negative b*) could be related to the second transition band of the octahedral Co(II) ion, ^4^T_1_ → ^4^A_2_. The greatest amounts of red colour (positive a*) and blue colour (negative b*) were obtained when the totality of Co(II) was almost entirely in the pentacoordinated site (x > 1.0), with a great distortion at 1200 °C (Table 3). The optimal compositions could be established when 2.0 ≤ x ≤ 2.5 (highest a* and lowest b*) at this temperature.

The colour parameters of the powdered Mg_x_Co_3−x_P_2_O_8_ (1.5 ≤ x ≤ 2.5) solid solutions in this study—with a* of 18.62 to 31.58 and b* of −18.33 to −24.17 (Table 5)—were, in absolute value, greater than those of the Co_x_Zn_2−x_SiO_4_ compositions (1.5 ≤ x ≤ 2.0 with the olivine structure and 0.05 ≤ x ≤ 1.00 with the willemite structure), which had a* of –9.3 to 4.4 and b* of −1.8 to −20.3 [15], and all of them were fired at 1200 °C. A greater amount of red colour (+a*) that was comparable with the greater amount of blue colour (-b*) was obtained in the solid solutions with the stable Co_3_P_2_O_8_ structure. So, the powdered Mg_2.5_Co_0.5_P_2_O_8_ composition fired at 1200 °C considerably reduced the amount of cobalt, keeping a colour comparable with that in the Co_3_P_2_O_8_ pigment, and its melting point was higher than 1200 °C. This composition could be used as a violet inorganic pigment in substitution for the Co_3_P_2_O_8_ inorganic pigment, thus decreasing its toxicity due to large amount of cobalt. Co_3_P_2_O_8_ is classified as a pigment (DCMA-8-11-1, DCMA Classification of Mixed Metal Oxide Inorganic Colour Pigments). Pigments include naturally occurring substances prepared from minerals or their combustion products, as well as synthetic compounds produced from appropriate raw materials. Pigments are insoluble in the surrounding media, and their optical effect arises from selective light absorption [19]. Therefore, the solid solutions prepared here are also inorganic pigments when unglazed.

Figure 8 shows the visible spectra in glazed tiles prepared with 3% Mg_x_Co_3−x_P_2_O_8_ (1.5 ≤ x ≤ 3.0) materials fired at 1200 °C. The bands assigned to Co(II) in the octahedral site and to Co(II) in a square-planar pyramid coordination were detected at higher wavelengths in the enamelled samples than in the powdered samples. This displacement increased the absorbance in the range of 593–650 nm and slightly decreased the absorbance at about 550 nm, so the colour observed in these enamelled materials was the characteristic cobalt blue colour obtained from purple powdered materials (1.5 ≤ x ≤ 2.5). The absorbance between 450 and 630 nm decreased in the enamelled Co_3_P_2_O_8_ composition with respect to the powdered Co_3_P_2_O_8_ at 1000 °C. This decrease was not detected in the Mg_2.0_Co_1.0_P_2_O_8_ and Mg_2.5_Co_0.5_P_2_O_8_ compositions that were fired at 1200 °C with their lower amount of cobalt. The violet colour of the powdered samples changed to the characteristic cobalt blue due to the change in the local environment of the Co(II) ions, which could be visualised in the UVV spectra of the glazed samples with the displacement of the bands towards higher wavelengths and with the appearance of a new band assigned to tetrahedral Co(II). This blue colour was also obtained with Co_2_SiO_4_, MgCoSiO_4_ or Co_3_P_2_O_8_ pigments containing a greater amount of cobalt.

The colour parameters (L* a* b*) of the enamelled samples under the conditions of this study with the commercial glaze used are included in Table 6. A dark blue colour was obtained from the compositions with 1.5 ≤ x ≤ 2.0, and a blue colour with a greater lightness was obtained from the composition with x = 2.5. The Mg_x_Co_3−x_P_2_O_8_ (1.5 ≤ x ≤ 2.5) solid solutions with the stable Co_3_P_2_O_8_ structure may be used as blue pigments in the ceramic industry. The CIE L*/a*/b* colour parameters of classical blue pigments used in the ceramic industry with Co_2_SiO_4_ or MgCoSiO_4_ compositions with an olivine structure (weight ratio of pigment to glaze equal to 1:5, 20 weight% pigment) were 29.00/11.20/–25.6 and 29.19/7.97/–17.61, respectively, for the single-fired enamelled samples [20,21]. The glazed tiles from the Mg_x_Co_3−x_P_2_O_8_ (1.5 ≤ x ≤ 2.5) solid solutions (including 3% pigment) showed blue colourations with a large amount of blue colour (–15.9 ≤ b* ≤ –20.20) and a low lightness (18.53 ≤ L* ≤ 27.05). The amount of cobalt in the compositions was between 28.1 weight% (x = 1.5) and 10.5 weight% (x = 2.5), while it was 56.1 weight% in Co_2_SiO_4_ and 33.6 weight% in MgCoSiO_4_. The use of the Mg_x_Co_3−x_P_2_O_8_ (1.5 ≤ x ≤ 2.5) solid solutions with the stable Co_3_P_2_O_8_ structure as blue pigments reduced the amount of cobalt used with respect to the amount of cobalt used in Co_2_SiO_4_ and MgCoSiO_4_ pigments because comparable values of blue (negative b*) were obtained with a smaller amount of cobalt in the composition of the pigment and with a pigment quantity that was lower by 6.7.

## 4. Conclusions

Mg_x_Co_3−x_P_2_O_8_ (0 ≤ x ≤ 3) solid solutions with the stable Co_3_P_2_O_8_ structure were synthesised via the chemical coprecipitation method. Their structural characterisation is consistent with the replacement of the Co(II) ion with the smaller Mg(II) ion. A decrease in the b unit cell parameter and the unit cell volume with x was obtained. The slight increase in the a and c unit cell parameters with x indicates that the Co_3_P_2_O_8_ structure is distorted when Mg(II) is incorporated into it. When x > 1.0, the decrease in the longest M-O distance with x is remarkable.

The experimental distribution of Co(II) between the two positions in the solid solutions with the stable Co_3_P_2_O_8_ structure indicates that the Co(II) ion presents a preference for the site with CN = 5 (4e) and the Mg(II) ion presents a preference for the octahedral position (2a). The bands in the UV–vis–NIR spectra can be assigned to Co(II) in octahedral and square-planar pyramid coordination. The most noticeable decrease in absorbance with composition in the spectra was due to the decrease in the total Co(II) in the compositions. A slight decrease in the absorbance from the first transition, which was assigned to Co(II) in the octahedral site, with respect to the transition assigned to Co(II) in a square-planar pyramid coordination was also detected.

Purple and violet colours were obtained when a stable Co_3_P_2_O_8_ structure was developed from Mg_x_Co_3−x_P_2_O_8_ (0 ≤ x ≤ 3) compositions. These colourations were kept at 1000 and 1200 °C when x ≥ 1.5, so inorganic pigments with a smaller amount of Co(II) were obtained. From powdered samples, optimal compositions can be established when 2.0 ≤ x ≤ 2.5 (highest a* and lowest b*) at this temperature. This highlights the purple colour with the values of a* = 31.6 and b* = –24.2 obtained from the powdered Mg_2.5_Co_0.5_P_2_O_8_ composition fired at 1200 °C, which minimised the amount of cobalt and improved both the thermal stability and the colour of the Co_3_P_2_O_8_ pigment. Mg_x_Co_3−x_P_2_O_8_ (2.0 ≤ x ≤ 2.5) solid solutions are considered the optimal compositions for obtaining the characteristic cobalt blue in glazed tiles.

## Figures and Tables

**Figure 1 materials-15-01111-f001:**
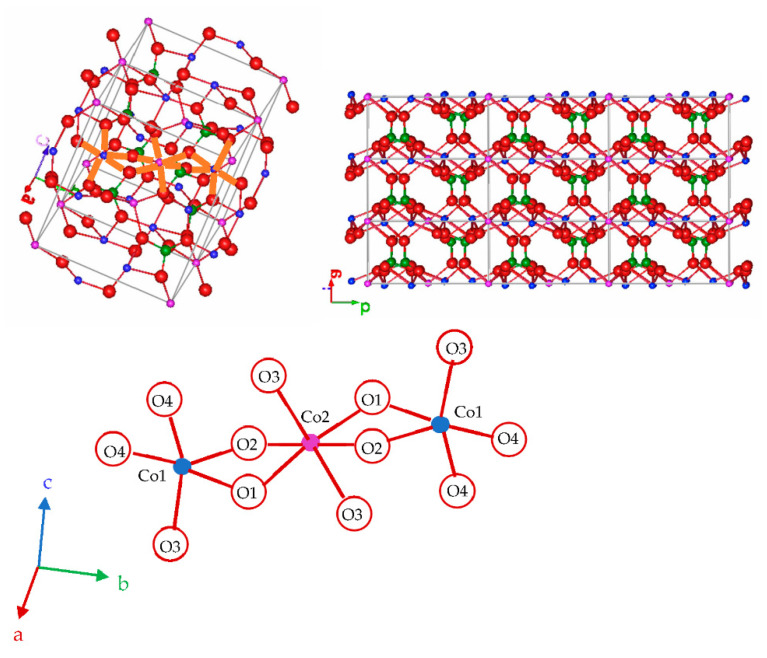
Stable Co_3_P_2_O_8_ structure. The colour scheme to represent the atoms is O (-II): red, P(V): green, Co(II): blue with CN = 5, and lilac with CN = 6.

**Figure 2 materials-15-01111-f002:**
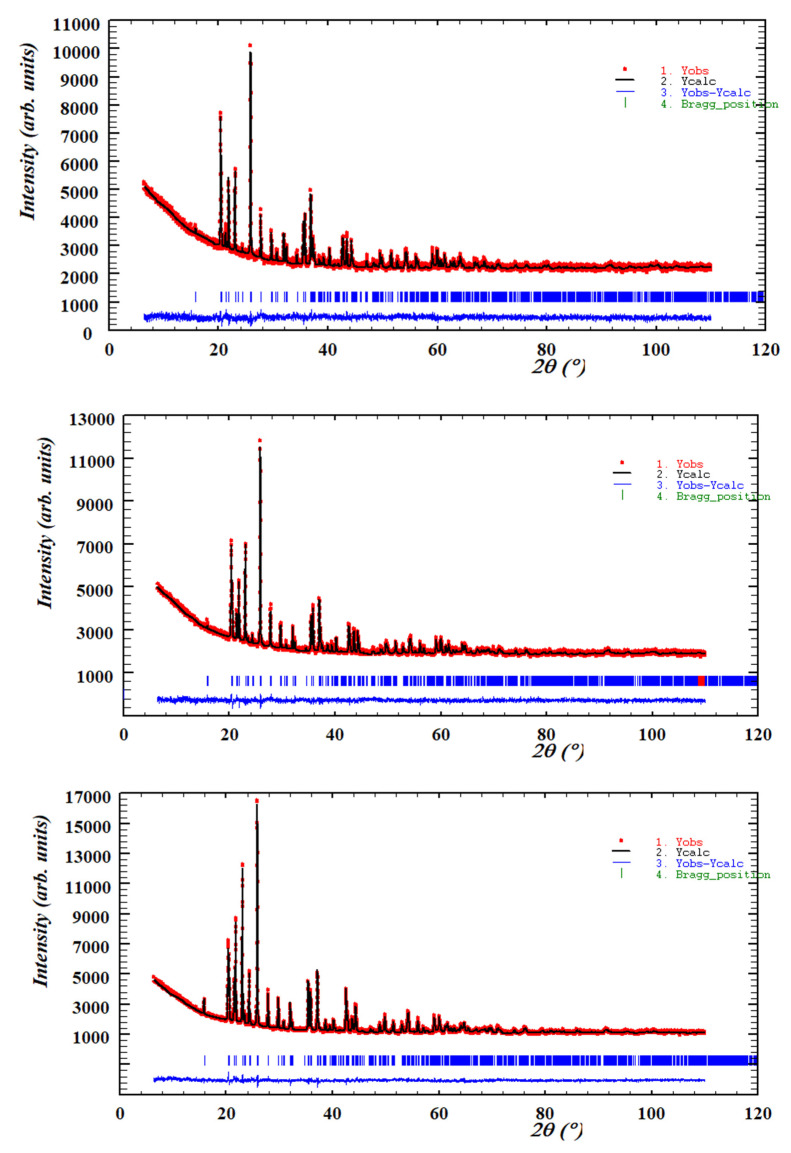
The diffraction profile refinement obtained with Rietveld’s method for the Mg_0.5_Co_2.5_P_2_O_8_ (x = 0.5), Mg_1.5_Co_1.5_P_2_O_8_ (x = 1.5) and Mg_2.5_Co_0.5_P_2_O_8_ (x = 2.5) compositions fired at 1000 °C.

**Figure 3 materials-15-01111-f003:**
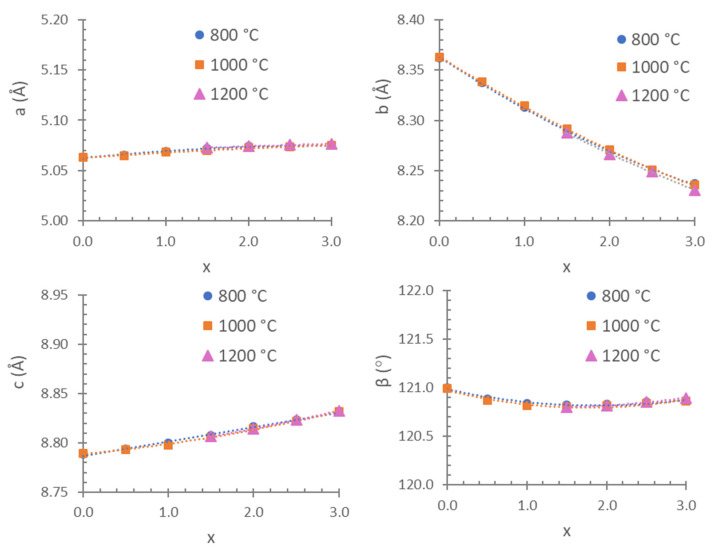
Unit cell parameters in the Co_3_P_2_O_8_ structure from the Mg_x_Co_3−x_P_2_O_8_ (0.0 ≤ x ≤ 3.0) compositions fired at 800, 1000 and 1200 °C.

**Figure 4 materials-15-01111-f004:**
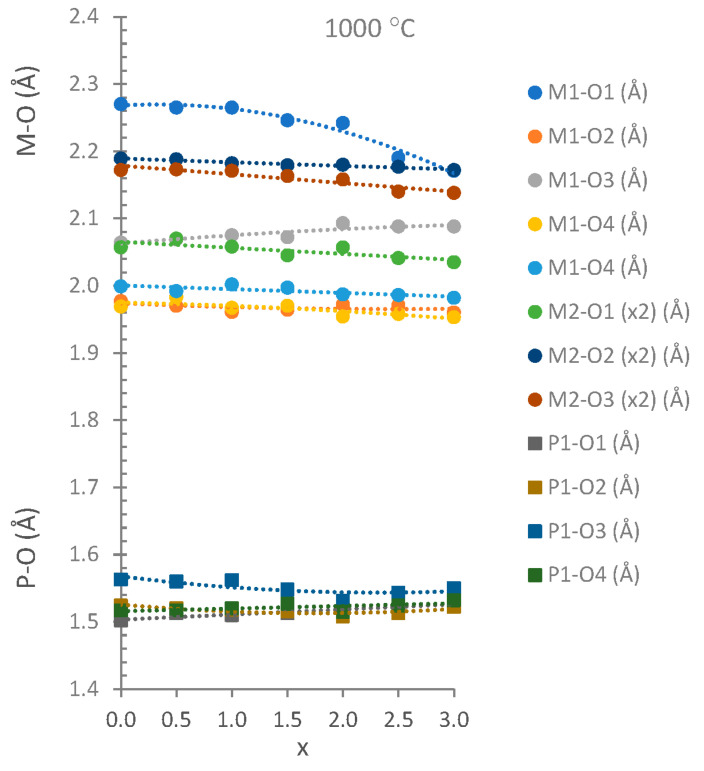
M-O (circles) and P-O (squares) distances in the stable Co_3_P_2_O_8_ structure at 1000 °C.

**Figure 5 materials-15-01111-f005:**
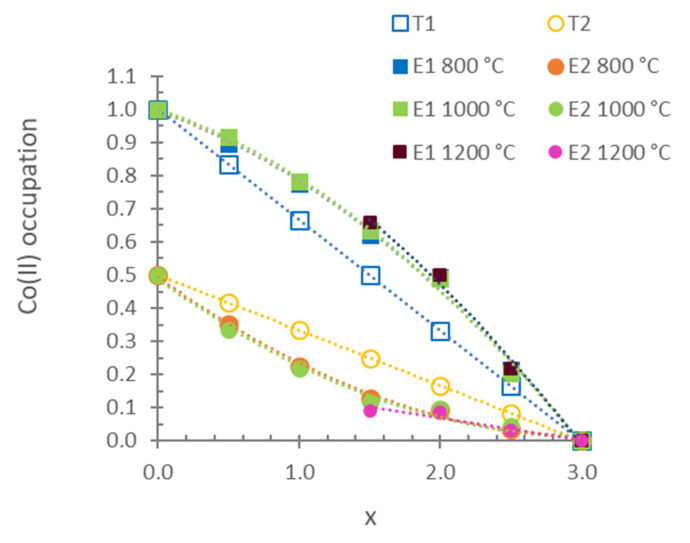
Variations in the occupation of Co(II) ions with composition in sites of the stable Co_3_P_2_O_8_ structure from the fired Mg_x_Co_3−x_P_2_O_8_ compositions. E1: experimental occupation in pentacoordinated sites (M1), E2: experimental occupation in octahedral sites (M2), T1: theoretical random occupation in M1 sites and T2: theoretical random occupation in M2 sites.

**Figure 6 materials-15-01111-f006:**
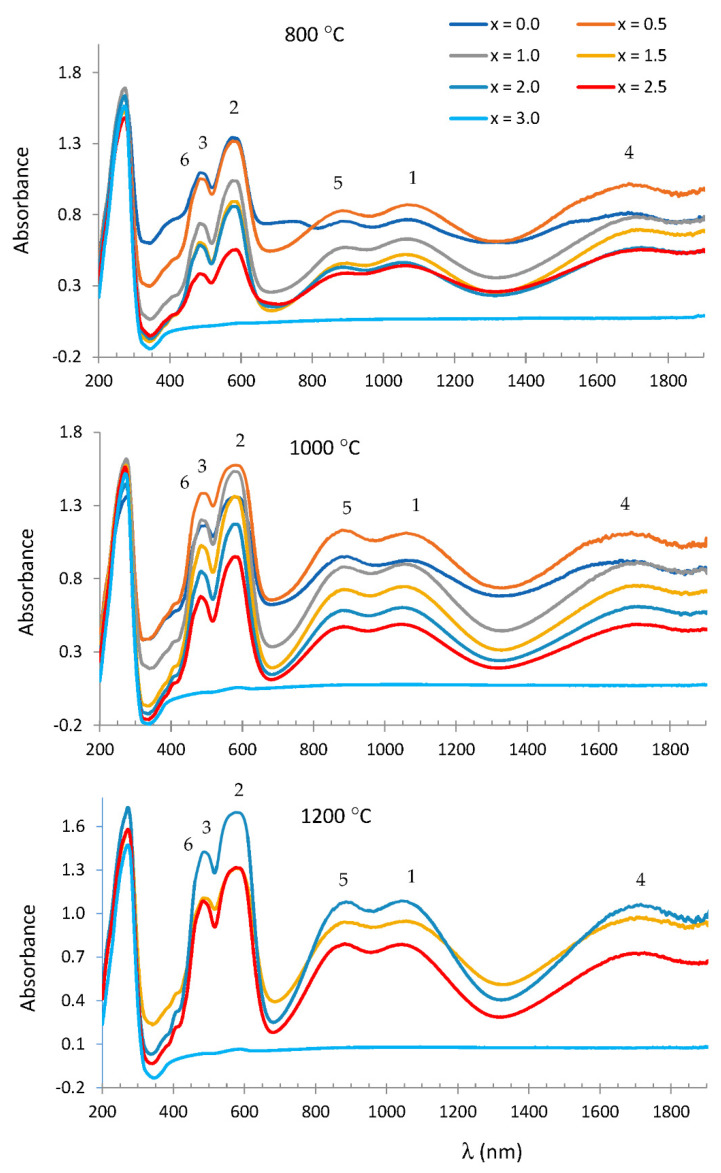
UV–vis–NIR spectra of Mg_x_Co_3−x_P_2_O_8_ fired at 800, 1000 and 1200 °C. Co(II) CN = 6: (1) ^4^T_1_ → ^4^T_2_, (2) ^4^T_1_ → ^4^A_2_, (3) ^4^T_1_ → ^4^T_1_(P). Co(II) CN = 5:, (4) ^4^A_2_ → ^4^A_1_, (5) ^4^A_2_ → ^4^E, (6) ^4^A_2_ → ^4^E(P).

**Figure 7 materials-15-01111-f007:**
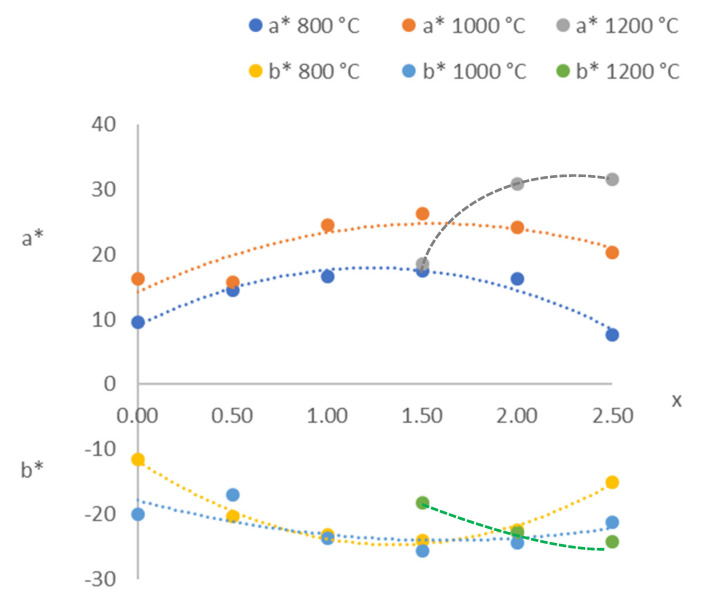
CIE L* a* b* from the Mg_x_Co_3−x_P_2_O_8_ compositions fired at 800, 1000 and 1200 °C.

**Figure 8 materials-15-01111-f008:**
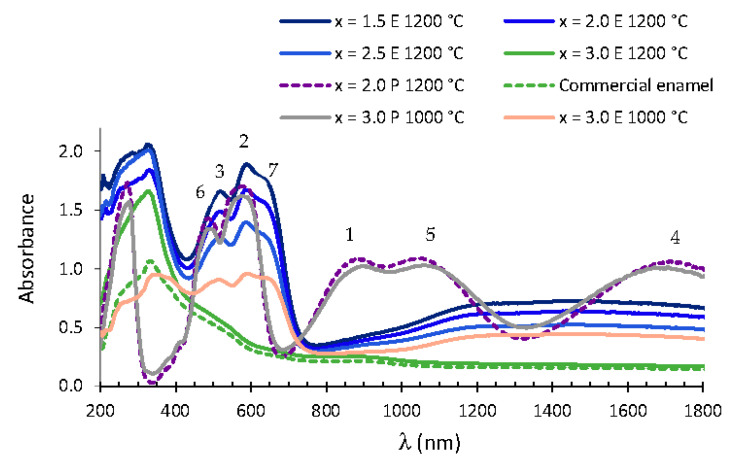
UV–vis–NIR spectra of 3% Mg_x_Co_3−x_P_2_O_8_ (0.0 ≤ x ≤ 3.0) compositions fired at 1200 °C and the commercial enamel and composition fired at 1000 °C with x = 3.0. E: enamelled sample. P = powdered sample. Co(II) CN= 6: (1) ^4^T_1_ → ^4^T_2_, (2) ^4^T_1_ → ^4^A_2_, (3) ^4^T_1_ → ^4^T_1_(P). Co(II) CN = 5: (4) ^4^A_2_ → ^4^A_1_, (5) ^4^A_2_ → ^4^E, (6) ^4^A_2_ → ^4^E(P). Co(II) CN = 4 (tetrahedral coordination): (7) ^4^A_2_ → ^4^T_1_(P).

**Table 1 materials-15-01111-t001:** Evolution of crystalline phases with temperature in the Mg_x_Co_3−x_P_2_O_8_ (0.0 ≤ x ≤ 3.0) compositions.

X	800 °C	1000 °C	1200 °C
0.0	C(s)	C(s)	
0.5	C(s)	C(s)	
1.0	C(s)	C(s)	
1.5	C(s)	C(s)	C(s)
2.0	C(s), M(vw)	C(s)	C(s)
2.5	C(s), M(vw)	C(s)	C(s)
3.0	C(s), M(w)	C(s)	C(s)

Crystalline phases: C = stable Co_3_P_2_O_8_, Mg_3_P_2_O_8_, or solid solutions with the same structure; M = Mg_2_P_2_O_7_. Diffraction peak intensity: s = strong, vw = very weak.

**Table 2 materials-15-01111-t002:** Variations in unit cell parameters in the stable Co_3_P_2_O_8_ structure obtained from fired Mg_x_Co_3−x_P_2_O_8_ (0.0 ≤ x ≤ 3.0) compositions.

T (°C)	x	a (Å)	b (Å)	c (Å)	β (°)	V (Å^3^)
800	0.0	5.0629(1)	8.3618(2)	8.7888(2)	121.003(1)	318.92(1)
800	0.5	5.0655(1)	8.3377(2)	8.7938(2)	120.882(1)	318.75(1)
800	1.0	5.0691(1)	8.3131(1)	8.7999(2)	120.840(1)	318.40(1)
800	1.5	5.07242(9)	8.2887(1)	8.8076(1)	120.8274(9)	318.018(9)
800	2.0	5.07424(8)	8.2691(1)	8.8169(1)	120.8303(8)	317.675(8)
800	2.5	5.0734(1)	8.2505(2)	8.8237(2)	120.854(1)	317.07(1)
800	3.0	5.07529(9)	8.2370(1)	8.8316(2)	120.862(1)	316.93(1)
1000	0.0	5.06288(8)	8.3631(1)	8.7894(1)	120.9897(8)	319.027(8)
1000	0.5	5.06491(9)	8.3386(1)	8.7933(2)	120.8665(9)	318.78(1)
1000	1.0	5.0677(1)	8.3143(2)	8.7981(2)	120.820(1)	318.35(1)
1000	1.5	5.07022(8)	8.2914(1)	8.8056(1)	120.7983(8)	317.975(8)
1000	2.0	5.07251(6)	8.2704(1)	8.8146(1)	120.813(7)	317.588(7)
1000	2.5	5.07328(5)	8.25084(9)	8.82223(9)	120.8395(5)	317.073(6)
1000	3.0	5.07521(4)	8.23586(8)	8.83144(7)	120.8663(5)	316.860(5)
1200	1.5	5.07299(7)	8.2876(1)	8.8068(1)	120.8024(8)	318.034(7)
1200	2.0	5.07408(4)	8.26660(7)	8.81422(8)	120.8155(5)	317.520(5)
1200	2.5	5.07550(3)	8.24883(5)	8.82347(5)	120.8519(4)	317.139(3)
1200	3.0	5.07675(2)	8.23098(4)	8.83270(4)	120.8969(3)	316.712(3)

**Table 3 materials-15-01111-t003:** Variations in the M-O (M = Co, Mg) distances in the stable Co_3_P_2_O_8_ structure with temperature in the Mg_x_Co_3−x_P_2_O_8_ compositions.

T (°C)	X	M1-O1 (Å)	M1-O2 (Å)	M1-O3 (Å)	M1-O4 (Å)	M1-O4 (Å)	M2-O1 (x2) (Å)	M2-O2 (x2) (Å)	M2-O3 (x2) (Å)
800	0.0	2.274(6)	1.973(5)	2.065(7)	1.960(5)	1.996(5)	2.052(8)	2.195(4)	2.174(4)
800	0.5	2.273(6)	1.970(5)	2.076(7)	1.985(5)	1.992(5)	2.089(7)	2.184(4)	2.173(4)
800	1.0	2.253(5)	1.969(5)	2.070(7)	1.969(4)	1.999(4)	2.059(7)	2.183(4)	2.164(4)
800	1.5	2.246(5)	1.972(4)	2.074(6)	1.961(4)	1.995(4)	2.054(6)	2.185(3)	2.155(4)
800	2.0	2.242(5)	1.979(4)	2.091(6)	1.941(4)	1.985(4)	2.061(6)	2.188(3)	2.153(4)
800	2.5	2.228(5)	1.967(4)	2.102(4)	1.946(4)	1.991(4)	2.050(5)	2.187(3)	2.150(3)
800	3.0	2.226(5)	1.950(5)	2.098(6)	1.941(4)	1.980(4)	2.058(5)	2.182(3)	2.153(3)
1000	0.0	2.270(6)	1.977(5)	2.064(7)	1.969(5)	1.999(4)	2.057(7)	2.189(4)	2.172(4)
1000	0.5	2.265(6)	1.970(5)	2.069(7)	1.983(4)	1.992(4)	2.070(7)	2.188(4)	2.173(4)
1000	1.0	2.265(6)	1.961(5)	2.075(7)	1.967(5)	2.002(5)	2.058(7)	2.182(4)	2.171(4)
1000	1.5	2.246(5)	1.964(4)	2.072(6)	1.970(4)	1.997(4)	2.045(6)	2.179(3)	2.163(4)
1000	2.0	2.242(5)	1.972(4)	2.093(6)	1.954(4)	1.987(4)	2.057(6)	2.180(3)	2.158(4)
1000	2.5	2.190(4)	1.971(4)	2.088(5)	1.958(3)	1.986(4)	2.041(4)	2.177(3)	2.140(3)
1000	3.0	2.171(3)	1.960(3)	2.088(4)	1.953(3)	1.982(3)	2.035(3)	2.172(2)	2.138(2)
1200	1.5	2.229(6)	1.979(5)	2.078(7)	1.950(5)	1.983(5)	2.036(8)	2.201(4)	2.157(4)
1200	2.0	2.231(5)	1.956(4)	2.067(6)	1.965(4)	2.009(4)	2.037(6)	2.190(3)	2.171(4)
1200	2.5	2.203(4)	1.968(4)	2.085(5)	1.961(3)	2.000(4)	2.042(4)	2.162(3)	2.151(3)
1200	3.0	2.166(3)	1.956(3)	2.081(4)	1.965(3)	1.988(3)	2.033(3)	2.172(2)	2.138(2)

**Table 4 materials-15-01111-t004:** Variations in the P-O distances in the stable Co_3_P_2_O_8_ structure with temperature from the Mg_x_Co_3−x_P_2_O_8_ compositions.

T (°C)	X	P1-O1 (Å)	P1-O2 (Å)	P1-O3 (Å)	P1-O4 (Å)
800	0.0	1.944(9)	1.527(5)	1.561(5)	1.512(5)
800	0.5	1.521(9)	1.520(5)	1.566(5)	1.517(5)
800	1.0	1.512(9)	1.526(5)	1.550(5)	1.527(5)
800	1.5	1.516(8)	1.521(4)	1.546(4)	1.519(4)
800	2.0	1.518(7)	1.508(4)	1.537(4)	1.521(4)
800	2.5	1.516(7)	1.507(4)	1.534(3)	1.515(4)
800	3.0	1.524(6)	1.520(4)	1.544(3)	1.527(4)
1000	0.0	1.502(9)	1.524(5)	1.563(5)	1.517(5)
1000	0.5	1.513(9)	1.520(5)	1.560(5)	1.517(5)
1000	1.0	1.510(9)	1.518(5)	1.562(5)	1.520(5)
1000	1.5	1.513(7)	1.516(4)	1.548(4)	1.527(4)
1000	2.0	1.514(7)	1.508(4)	1.531(4)	1.515(4)
1000	2.5	1.519(5)	1.513(3)	1.543(3)	1.524(3)
1000	3.0	1.532(4)	1.522(3)	1.550(3)	1.532(3)
1200	1.5	1.516(9)	1.526(5)	1.522(5)	1.520(5)
1200	2.0	1.520(7)	1.483(4)	1.549(4)	1.526(4)
1200	2.5	1.505(5)	1.511(3)	1.537(3)	1.548(3)
1200	3.0	1.530(4)	1.516(3)	1.549(3)	1.527(3)

**Table 5 materials-15-01111-t005:** CIE L*a*b* parameters of the Mg_x_Co_3−x_P_2_O_8_ compositions.

x.		Raw Material	300 °C	600 °C	800 °C	1000 °C	1200 °C
0.0	L*	51.36	32.62	35.23	36.73	31.07	
	a*	7.72	8.62	7.64	9.52	16.22	
	b*	–24.72	–11.72	–10.64	−11.50	−20.05	
	Colour	Violet	Purple	Dark violet	Dark violet	Purple	
0.5	L*	49.71	31.21	34.54	37.07	29.30	
	a*	6.00	7.88	10.72	14.57	15.72	
	b*	−30.36	–18.94	−22.32	−20.45	−16.98	
	Colour	Violet	Purple	Purple	Purple	Purple	
1.0	L*	37.31	44.69	51.37	52.43	35.18	
	a*	7.78	4.06	12.73	16.58	24.48	
	b*	–29.41	−27.07	−25.60	−23.25	−23.78	
	Colour	Violet	Violet	Violet	Violet	Purple	
1.5	L*	47.92	48.86	60.12	58.20	41.39	36.24
	a*	5.35	−2.54	15.05	17.52	26.21	18.62
	b*	−20.70	−38.37	−23.98	−24.11	−25.70	−18.33
	Colour	Violet	Blue	Violet	Violet	Violet	Purple
2.0	L*	40.20	60.85	59.14	58.97	49.49	31.33
	a*	−1.83	−10.40	14.03	16.26	24.25	30.93
	b*	−41.74	−30.24	−23.17	−22.42	−24.39	−22.79
	Colour	Blue	Blue	Violet	Violet	Violet	Purple
2.5	L*	54.41	64.37	68.93	68.73	56.60	38.86
	a*	−2.81	−24.25	3.77	7.54	20.30	31.58
	b*	−36.03	−29.37	−14.87	−15.13	−21.20	−24.17
	Colour	Blue	Light blue	Lilac	Lilac	Violet	Purple
3.0	L*	93.89	93.31	95.35	95.38	94.90	94.24
	a*	−0.10	−0.04	−0.15	−0.10	0.29	0.35
	b*	+0.38	+1.12	0.16	0.05	−0.48	−0.54
	Colour	White	White	White	White	White	White

**Table 6 materials-15-01111-t006:** CIE L* a* b* colour parameters from the glazed tiles obtained from the Mg_x_Co_3−x_P_2_O_8_ (1.5 _≤_ x _≤_ 3.0) materials fired at 1200 °C.

x	L*	a*	b*	Observed Colour
1.5	18.53	−0.24	−26.18	Dark blue
2.0	20.56	+0.00	−20.20	Dark blue
2.5	27.05	+0.55	−15.95	Blue
3.0	68.24	+13.22	+26.94	Beige

## Data Availability

Data cited in this study can be found in: https://icsd.fiz-karlsruhe.de/.

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
