# Peer review of "Cobalt Minimisation in Violet Co3P2O8 Pigment"

_materials, 2022, doi:10.3390/ma15031111_

Round 1

Reviewer 1 Report

The authors’ idea is good, and they wants to minimize the toxic and expensive cobalt amount in violet Co3P2O8 pigment. However, it is necessary to improve the quality of the manuscript comprehensively.

  1. The language and wording of the paper has to be carefully checked.
  2. The novelty of this research has not been specified.
  3. Research gaps have not been well identified. How is the literature review, the comparison between the results of the previous studies, where the gap issues will be solved?
  4. Rationality of experimental parameters (such as pH = 10) should offer.
  5. The yield from the research has not been calculated to ensure this method is efficient.

6. Physical comparison diagrams of pigments and glazing results are firmly recommended to offer.

Author Response

  1. The text has been revised.
  2. This study considers the limitations of the Cobalt violet orthophosphate, Co3P2O8, as a ceramic pigment because of the melting point of this compound is lower than the work temperature in the most of the products. MgxCo3-xP2O8 (0 ≤ x ≤ 3) solid solutions with the stable Co3P2O8 structure were synthesized via the chemical co-precipitation method. The formation of solid solutions between the isostructural Co3P2O8 and Mg3P2O8 compounds increases the melting point at temperature higher than 1200 °C when x ≥ 1.5 and decreases the toxic high amount of cobalt in this pigment. The substitution of Co(II) by Mg(II) decreases the toxicity of these materials and decreases its price, hence the interest of these materials in the ceramic industry is higher.

        As far as we know, no information about the MgxCo3-xP2O8 solid solutions
        at T ≥1000 °C has been reported. 

  1. Structural information about the MgxCo3-xP2O8 solid solutions at 800 °C was published by Nord et al. 1980, Ref. 13. Values of the unit cell parameters in bibliography at 800 °C have not been added to the Figure 3 together to the values obtained in this study at 800, 1000 and 1200 °C because we have used the std cell that is not directly comparable with the published cell.
  2. The text “Experimental parameters were chosen to obtain precipitates of the cations before drying the material. pH = 10 was chosen because although the Co(OH)2 precipitate at pH > 7, the Mg(OH)2 precipitate at pH > 9.5.” has been included in Experimental section.
  3. Colour of the materials with 2.0 ≤ x ≤ 2.5 at 1200 °C is the main result obtained in this study. CIE L* a* b* colour parameters were measured. No theoretical calculations have been necessary.
  4. Spectra and CIE L* a* b* colour parameters of pigments (powder materials) and enamelled samples with a commercial glaze are included in the article.

Reviewer 2 Report

Peer review in file.

Author Response

  1. Colour of some curves has been changed in Figure 6.
  2. From the results of this study, melting point is less than 1200 °C when 0 ≤ x ≤ 1.0 and it is higher than 1200 °C when 1.5 ≤ x ≤ 3.0. The increase of the Mg(II) amount in MgxCo3-xP2O8 solid solutions increases the thermal stability.
  3. Figure 2 includes the diffraction profile refinement by Rietveld´s method from Mg0.5Co2.5P2O8 (x = 0.5), Mg1.5Co1.5P2O8 (x = 1.5), and Mg2.5Co0.5P2O8 (x = 2.5) compositions fired at 1000 °C. XRD patterns are similar at 1200 °C than at 1000 °C in compositions with x > 1.5. Compositions with x < 1.5 melted at 1200 °C and their XRD patterns could not be obtained.

The Figure 3 is superimposed on the Figure 2 in the actual pdf file and only the caption of Figure 2 is visible. Figures 2 and 3 and their captions can be observed in the file with word format.

We regret not having detected the error in the current pdf file.

Reviewer 3 Report

In this paper, MgxCo3-xP2O8 (0≤x≤3) solid solutions with the stable Co3P2O8 structure were synthesized via the chemical co-precipitation method. The overall work is logical, but it lacks obvious innovation and highlights, this paper needs major revision. Therefore, there are some questions listed below for the authors’ consideration:

1.Part of the introduction should be improved. In the background section, the manuscript should not write too many words, and should be the focus on innovative points.

2.In “ Introduction”, the authors are encouraged to refer to the following publications :

Thermal Insulation Performance of Novel Coated Fabrics Based on Fe-Doped BaSnO3 Near-Infrared Reflectance Pigments, ACS Sustainable Chem. Eng. 9 (2021) 16328−16337.

3.In “Abstract”, the author holds “These compositions may be used as blue ceramic pigments.”. But the color of pigments in the title is violet. Hope the author will give a reasonable explanation

4.Please indicate the reference substance used in the UV- vis-NIR spectroscopy.

5.Figure 2 is wrong.

6.where is Figure 3?

7.How to determine the thermal stability?

8.why the Co-O distance (1.940 Å) is shorter than the Mg-O distance (1.969 Å)?

9.The article innovation is not obvious.

10.Format problem.

a)Unified font format.

b)page6 line 201, at 800, 1000?

c) Table 3, from

 MgxCo3-xP2O8?

e) The abscissa title should be placed in the middle.

Author Response

  1. Text in introduction has been changed.
  2. The authors have not considered adequate the incorporation of this reference in the introduction. This article is not related with our study. The structure and doping cation are different in them.
  3. Abstract. The text “The violet colour of powder samples changes to characteristic cobalt blue because the bands assigned to Co(II) in spectra are detected at higher wavelength in enamelling samples than in powder samples.” has been included before “These compositions may be used as blue ceramic pigments”.
  4. The text “The ultraviolet visible near infrared (UV-vis-NIR) spectra in the 200 to 2500 nm range were obtained using a Jasco V-670 spectrophotometer” has been changed to “The ultraviolet visible near infrared (UV-vis-NIR) spectra in the 200 to 2500 nm range were obtained using a Jasco V-670 spectrophotometer and BaSO4 as reference substance”.
  5. Figure 2 includes the diffraction profile refinement by Rietveld´s method from Mg0.5Co2.5P2O8 (x = 0.5), Mg1.5Co1.5P2O8 (x = 1.5), and Mg2.5Co0.5P2O8 (x = 2.5) compositions fired at 1000 °C. Figure 2 and its caption can be observed in the file with word format.
  6. The Figure 3 is superimposed on the Figure 2 in the current pdf file and only the caption of Figure 2 is visible. Figure 3 and its caption can be observed in the file with word format.

         We regret not having detected the error in the current pdf file.

  1. In this study compositions were characterized at different temperatures. From the results of this study, melting point is less than 1200 °C when 0 ≤ x ≤ 1.0 and it is higher than 1200 °C when 1.5 ≤ x ≤ 3.0. The increase of the Mg(II) amount with the formation of MgxCo3-xP2O8 solid solutions increases the stability at high temperature (thermal stability).
  2. Differences in electronic configuration with the seven electrons in 3d orbitals in Co(II) ion with small orbital penetration effect could explain the shorter Co-O distance (1.940 Å) than the Mg-O distance (1.969 Å) although the ionic radius is bigger in Co(II) ion than in Mg(II) ion. The higher covalence in Co-O bond than in Mg-O bond could explain the shorter Co-O distances.
  3. This study considers the limitations of the Cobalt violet orthophosphate, Co3P2O8, as a ceramic pigment because of the melting point of this compound is lower than the work temperature in the most of the products. MgxCo3-xP2O8 (0 ≤ x ≤ 3) solid solutions with the stable Co3P2O8 structure were synthesized via the chemical co-precipitation method. The formation of solid solutions between the isostructural Co3P2O8 and Mg3P2O8 compounds increases the melting point at temperature higher than 1200 °C when x ≥ 1.5 and decreases the toxic high amount of cobalt in this pigment. The substitution of Co(II) by Mg(II) decreases the toxicity of these materials and decreases its price, hence the interest of these materials in the ceramic industry is higher.

         Information about the MgxCo3-xP2O8 solid solutions published does not             include information at T ≥1000 °C.

  1. a) Font format has been revised.
  2. b) The Figure 3 is superimposed on the Figure 2 in the current pdf file and only the caption of Figure 2 is visible. The text in these Figure captions is:

         Figure 2 The diffraction profile refinement by Rietveld´s method from                   Mg0.5Co2.5P2O8 (x = 0.5), Mg1.5Co1.5P2O8 (x = 1.5), and Mg2.5Co0.5P2O8             (x = 2.5) compositions fired at 1000 °C.

         Figure 3. Unit cell parameters in Co3P2O8 structure from MgxCo3-xP2O8               (0.0 ≤ x ≤ 3.0) compositions fired at 800, 1000 and 1200 °C.

  1. c) “Table 3: Variation of the M-O (M = Co, Mg) distances in stable Co3P2O8 structure with temperature from MgxCo3-xP2O8 compositions” has been changed to “Table 3: Variation of the M-O (M = Co, Mg) distances in stable Co3P2O8 structure with temperature in MgxCo3-xP2O8 compositions”.
  2. d) The abscissa title has been placed in the middle.

Reviewer 4 Report

In this paper, the MgxCo3-xP2O8 (0≤x≤3) solid solutions with stable Co3P2O8 structure were synthesized by the chemical co-precipitation method, it not only can enhance the thermal stability of the pigments, but also can decrease the toxic element of Co. But part of the existing problems in the paper are as follows:

  • The word “stables” in abstract should be “stable”, in the line 47 of the page 2, “decrease” should be corrected, in the line 54 of page 2, “increasing” should be “increase”, In page 2, the sentence of “In these compositions, the red amount is high at 1000 °C, but it decreases with x at 1200°C” is unstandardized, in the line 142 of page 4, “although small amounts of Co2P2O7 are also………”, but M=Mg2P2O7. There are some troubles in the sentence “It is possible to reduce the cobalt amount used from the Co2SiO4 and MgCoSiO4 pigments by the use of the MgxCo3-xP2O8 (1.5≤x≤2.5) solid solutions with the stable Co3P2O8 structure as blue ceramic pigments”. In conclusion part, in the line 386, “has” should be “have”. Besides, other grammatical mistakes are also supposed to be checked carefully in the manuscript.
  • The problems of figure format, the matched Co, P, and O should be marked in Figure 1. In Figure 4, Figure 5, Figure 6 and Figure 8, the annotation should be centered.
  • For the composition analysis, especially MgxCo3-xP2O8 compositions, it is essential to provide X-ray diffraction (XRD) pattern for readers to study the phase change at different doping amounts and calcinating temperatures.
  • How to verify the formation of these solid solutions increases the thermal stability. It is necessary to provide the TG-DSC measurement to illustrate this view.
  • It is rigorous to cite references at different formats in the references part, and you’d better to correct the format according to the requirement of this journal.

Author Response

  • Abstract: the word “stables” has been changed to “stable”.
  • Line 47, page 2: the word “decresase” has been changed to “decreases”.
  • Page 2: the text “The incorporation of Ni(II) in solid solutions with the stable Co3P2O8 structure decreases the blue amount. In these compositions, the red amount is high at 1000 °C, but it decreases with x at 1200 °C” has been changed to “The red amount is high when x > 1.0 at 1000 ° The incorporation of Ni(II) in solid solutions with the stable Co3P2O8 structure decreases the blue amount”.
  • Line 142, page 4: The text “although small amounts of Co2P2O7 are also ….” has been changed to “although small amounts of Mg2P2O7 are also ….”.
  • The text “It is possible to reduce the cobalt amount used from the Co2SiO4 and MgCoSiO4 pigments by the use of the MgxCo3-xP2O8 (1.5 ≤ x ≤ 2.5) solid solutions with the stable Co3P2O8 structure as blue ceramic pigments.” has been changed to “The use of the MgxCo3-xP2O8 (1.5 ≤ x ≤ 2.5) solid solutions with the stable Co3P2O8 structure as blue ceramic pigments reduce the cobalt amount used respect to the cobalt amount used from Co2SiO4 and MgCoSiO4 pigments because comparable values of blue (negative b*) are obtained with less cobalt amount in the composition of the pigment and with 6.7 less pigment quantity.”
  • Line 386: The word “has” has been changed by “have”.
  • Other grammatical mistakes have also been corrected
  • Figure 1: Co1-O and Co2-O bonds in a CoO5-CoO6-CoO5 group have been marked in the two represented unit cells.
  • Figure captions have been centred (Figures 4, 5, 6 and 8).
  • Figure 2 includes the diffraction profile refinement by Rietveld´s method from Mg5Co2.5P2O8 (x = 0.5), Mg1.5Co1.5P2O8 (x = 1.5), and Mg2.5Co0.5P2O8 (x = 2.5) compositions fired at 1000 °C. XRD patterns are similar at 1200 °C than at 1000 °C in compositions with x > 1.5. Compositions with x < 1.5 melted at 1200 °C and their XRD patterns could not be obtained.

The Figure 3 is superimposed on the Figure 2 in the current pdf file and only the caption of Figure 2 is visible. This error was not in submitted files (word and pdf). Figures 2 and 3 and their captions can be observed in the file with word format.

We regret not having detected the error in the actual pdf file.

  • From the results of this study, melting point is less than 1200 °C when 0 ≤ x ≤0 and it is higher than 1200 °C when 1.5 ≤ x ≤ 3.0. To obtain the exact value of the melting point of these solid solutions is not considered necessary in this study. So, the authors do not consider necessary to provide the TG-DSC measurement to verify that the increase of the Mg(II) amount with the formation of MgxCo3-xP2O8 solid solutions increases the thermal stability.
  • References: Format has been revised.

Reviewer 5 Report

Line 82: FPStudio program  

Line 94: Figure 1: Indicate in the figure legend the color scheme used to represent the atoms. For example, red: oxygen; blue, green, purple, etc.  

Line 153 and 154: According to the text, Figure 2 should present the XRD patterns, with Rietveld refinements. However, Figure 2 shows the unit cell parameters, etc, duplicating the results presented in Table 2. Not a single XRD pattern appears in the entire article, which deals with structural characterization.  

Line 242: In Figure 4, the authors could change the P-O bond data symbols from circles to squares, for example.  

Line 273: In Figure 6, the legends of the UV-Vis-NIR bands, 1, 2, 3, etc., are partially hidden by the lines of the graph.   Table 5: Line 297: Table 5, change "Darck" to dark.

Author Response

  • Line 82. “Studio program” has been changed by “FPStudio program”.
  • Line 94. “Figure 1: Stable Co3P2O8 structure” has been changed by “Figure 1: Stable Co3P2O8 The colour scheme to represent the atoms is: O(-II) red, P(V) green, Co(II) blue with CN = 5 and lilac with CN = 6”.
  • Lines 153 and 154. The Figure 3 is superimposed on the Figure 2 in the current pdf file and only the caption of Figure 2 is visible. This error was not in submitted files (word and pdf). Figures 2 and 3 and their captions can be observed in the file with word format.

Representation in Figure 3 of the values of unit cell parameters with the variation of the composition (included in Table 2) allows visualizing the deviation of linearity (Vegard’s law).

Figure 2 includes the diffraction profile refinement by Rietveld´s method from Mg0.5Co2.5P2O8 (x = 0.5), Mg1.5Co1.5P2O8 (x = 1.5), and Mg2.5Co0.5P2O8 (x = 2.5) compositions fired at 1000 °C.

We regret not having detected the error in the current pdf file.

  • Line 242. In Figure 4, circles in P-O distances have been changed to squares.
  • Line 273. In Figure 6, numbers 1, 2, 3, … have been whole showed.
  • Line 297. In Table 5 “Darck” has been changed to “Dark”.

Round 2

Reviewer 1 Report

The revised manuscript is improved for publication.

Author Response

The text has been revised according to the referee´s suggestions.

Reviewer 2 Report

The review is in the attached file.

Author Response

The text has been revised according to the referee´s suggestions. See attached file.

Reviewer 3 Report

Most of the comments have been replied, I recommend the acceptance of this article.

Author Response

(The authors gave the same response as above.)

Round 3

Reviewer 2 Report

I accept the comments of the authors of the manuscript.